# TASK-INDUCED REPRESENTATION LEARNING

**Jun Yamada**[1]*, **Karl Pertsch**[2], **Anisha Gunjal**[2], **Joseph J. Lim**[3]*[4]†
[1] University of Oxford, [2] University of Southern California,
[3] Korea Advanced Institute of Science and Technology, [4] Naver AI Lab

## ABSTRACT

In this work, we evaluate the effectiveness of representation learning approaches for decision making in visually complex environments. Representation learning is essential for effective reinforcement learning (RL) from high-dimensional inputs. Unsupervised representation learning approaches based on reconstruction, prediction or contrastive learning have shown substantial learning efficiency gains. Yet, they have mostly been evaluated in clean laboratory or simulated settings. In contrast, real environments are visually complex and contain substantial amounts of clutter and distractors. Unsupervised representations will learn to model such distractors, potentially impairing the agent's learning efficiency. In contrast, an alternative class of approaches, which we call *task-induced representation learning*, leverages task information such as rewards or demonstrations from prior tasks to focus on task-relevant parts of the scene and ignore distractors. We investigate the effectiveness of unsupervised and task-induced representation learning approaches on four visually complex environments, from Distracting DMControl to the CARLA driving simulator. For both, RL and imitation learning, we find that representation learning generally improves sample efficiency on unseen tasks even in visually complex scenes and that task-induced representations can double learning efficiency compared to unsupervised alternatives. [1]

## 1 INTRODUCTION

The ability to compress sensory inputs into a compact representation is crucial for effective decision making. Deep reinforcement learning (RL) has enabled the learning of such representations via end-to-end policy training (Mnih et al., 2015; Levine et al., 2016). However, learning representations from reward feedback requires many environment interactions. Instead, *unsupervised* objectives for representation learning (Lange and Riedmiller, 2010; Finn et al., 2016; Hafner et al., 2019; Oord et al., 2018; Chen et al., 2020) directly maximize lower bounds on the mutual information between the sensory inputs and the learned representation. Empirically, such unsupervised objectives can accelerate the training of deep RL agents substantially (Laskin et al., 2020; Stooke et al., 2021; Yang and Nachum, 2021). However, prior works have evaluated such representation learning approaches in clean laboratory or simulated settings (Finn et al., 2016; Lee et al., 2020; Laskin et al., 2020) where most of the perceived information is important for the task at hand. In contrast, realistic environments feature lots of task-irrelevant detail, such as clutter or objects moving in the background. This has the potential to impair the performance of unsupervised representations for RL since they aim to model *all* information in the input equally.

The goal of this paper is to evaluate the effectiveness of representation learning approaches for decision making in such more realistic, visually complex environments with distractors and clutter. We evaluate two classes of representation learning approaches: (1) unsupervised representation learning based on reconstruction, prediction or contrastive objectives, and (2) approaches that leverage task information from prior tasks to shape the representation, which we will refer to as *task-induced representation learning* approaches. Such task information can often be available at no extra cost, e.g., in the form of prior task rewards if the data has been collected from RL training runs. Task-induced representations can leverage this task information to determine which features are important in visually complex scenes, and which are distractors that can be ignored. We focus

---

*Work done while at USC   † AI Advisor at NAVER AI Lab
[1]Project website: clvrai.com/tarp

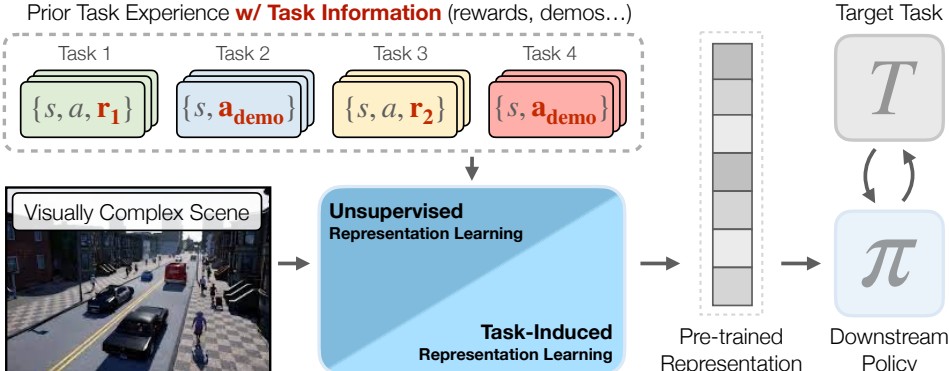

Figure 1: Overview of our pipeline for evaluating representation learning in visually complex scenes: given a multi-task dataset of prior experience we pre-train representations using unsupervised objectives, such as prediction and contrastive learning, or task-induced approaches, which leverage task-information from prior tasks to learn to focus on task-relevant aspects of the scene. We then evaluate the efficiency of the pre-trained representations for learning unseen tasks.

our evaluation on a practical representation learning setting, shown in Figure 1: given a large offline dataset of experience collected across multiple prior tasks, we perform offline representation learning with different approaches. Then, we transfer the pre-trained representations for training a policy on a new task and compare efficiency to training the policy from scratch.

To evaluate the effectiveness of different representation learning objectives, we compare common unsupervised approaches based on reconstruction, prediction and contrastive learning to four task-induced representation learning methods. We perform experiments in four visually complex environments, ranging from scenes with distracting background videos in the Distracting DMControl bench (Stone et al., 2021) to realistic distractors such as clouds, weather and urban scenery in the CARLA driving simulator (Dosovitskiy et al., 2017). Across RL and imitation learning experiments we find that pre-trained representations accelerate downstream task learning, even in visually complex environments. Additionally, in our comparisons task-induced representation learning approaches achieve up to double the learning efficiency of unsupervised approaches by learning to ignore distractors and task-irrelevant visual details.

In summary, the contributions of our work are threefold: (1) we formalize the class of *task-induced representation learning* approaches which leverage task-information from prior tasks for shaping the learned representations, (2) we empirically compare widely used unsupervised representation learning approaches and task-induced representation learning methods across four visually complex environments with numerous distractors, showing that task-induced representations can lead to more effective learning and (3) through analysis experiments we develop a set of best practices for task-induced representation learning.

## 2 RELATED WORK

Deep RL can train **"end-to-end" policies** that directly map raw visual inputs to action commands (Mnih et al., 2015; Kalashnikov et al., 2018). To improve data efficiency in deep RL from high-dimensional observations, a number of recent works have explored using data augmentation (Yarats et al., 2021; Laskin et al., 2021) and **unsupervised representation learning** techniques during RL training. The latter can be categorized into (1) reconstruction (Lange and Riedmiller, 2010; Finn et al., 2016), (2) prediction (Hafner et al., 2019; Lee et al., 2020), and (3) contrastive learning (Laskin et al., 2020; Stooke et al., 2021; Zhan et al., 2020; Yang and Nachum, 2021) approaches. While these works have shown improved sample efficiency, fundamentally they are *unsupervised* and therefore cannot decide which information in the input is relevant or irrelevant to a certain task. We show that this leads to deteriorating performance in visually complex environments such as autonomous driving scenarios with lots of non-relevant information in the input observations.

Another line of work has used task information to shape representations that learn to ignore distractors. Specifically, Fu et al. (2021) predict future rewards to learn representations that focus only on task-

relevant aspects of a scene. Zhang et al. (2021) leverage rewards as part of a bisimulation objective to learn representations without reconstruction. In this work, we provide a unifying framework for such task-induced representation learning approaches that encompasses the objectives of Fu et al. (2021) and Zhang et al. (2021) among others, and perform a systematic comparison to unsupervised representations in visually complex environments.

Closest to our work is the work of Yang and Nachum (2021), which evaluates unsupervised and task-induced representation learning objectives for pre-training in multiple environments from the D4RL benchmark (Fu et al., 2020). Their work shows that representation pre-training can substantially improve downstream learning efficiency. While Yang and Nachum (2021)'s evaluation is performed in visually clean OpenAI gym environments (Brockman et al., 2016) without distractors, we focus our evaluation on more realistic, visually complex environments with distractors, which can pose additional challenges, especially for unsupervised representation learning approaches.

## 3 PROBLEM FORMULATION

Our goal is to efficiently learn a policy $\pi$ that solves a target task $T_{\text{target}}$ in an MDP defined by a tuple $(\mathcal{S}, \mathcal{A}, \mathcal{T}, R, \rho, \gamma)$ of states, actions, transition probabilities, rewards, initial state distribution, and discount factor. The policy is trained to maximize the discounted return $\mathbb{E}_{a_t \sim \pi} \left[ \sum_{t=0}^{T-1} \gamma^t R(s_t, a_t) \right]$.

We do not assume access to the underlying state $s$ of the MDP, but instead our policy receives a high-dimensional observation $x \in \mathcal{X}$ in every step, e.g., an image observation. To improve training efficiency, we aim to learn an encoder $\phi(x)$ which maps the input observation to a low-dimensional *state representation* that is input to the policy $\pi(\phi(x))$. To learn this representation, we assume access to a dataset of past interactions $\mathcal{D} = \{\mathcal{D}_1, \ldots, \mathcal{D}_N\}$ collected across $T_{1:N} \in \mathcal{T}$ tasks, with per-task datasets $\mathcal{D}_i = \{x_t, a_t, (r_t), ...\}$ of state-action trajectories and optional reward annotation. The set of training tasks does not include the target task $T_{\text{target}} \notin T_{1:N}$. We assume that the training data contains task information for prior tasks, e.g., in the form of step-wise reward annotations $r_t$ or by consisting of task demonstrations. Such task information can often be available at no extra cost, e.g., if the data was collected from prior training runs (Fu et al., 2020; Çaglar Gülçehre et al., 2020) or human teleoperation (Mandlekar et al., 2018; Cabi et al., 2019).

We follow Stooke et al. (2021) and test all representation learning approaches in two phases: we first train the representation encoder $\phi(x)$ from the offline dataset $\mathcal{D}$, then transfer its parameters to the policy $\pi(\phi(x))$ and optimize it on the downstream task, either freezing or finetuning the encoder parameters. This allows us to efficiently reuse data collected across prior tasks.

### 3.1 UNSUPERVISED REPRESENTATION LEARNING

Unsupervised representation learning approaches aim to learn the low-dimensional representation $\phi(x)$, by maximizing lower bounds on the mutual information $\max_\phi I(x, \phi(x))$.

**Generative Representation Learning.** Maximizes the mutual information via generative modeling, either of the current state (**reconstruction**) or future states (**prediction**). A decoder $D(\phi(x))$ is trained by maximizing a lower bound on the data likelihood $p(x)$ (Kingma and Welling, 2014; Rezende et al., 2014; Denton and Fergus, 2018; Hafner et al., 2019).

**Contrastive Representation Learning.** Avoids training complex generative models by using contrastive objectives for maximizing the mutual information (Oord et al., 2018; Laskin et al., 2020; Stooke et al., 2021). During training, mini-batches of positive and negative data pairs are assembled, where positive pairs can for example constitute consecutive frames of the same trajectory while negative pairs mix frames across trajectories. The representation is then trained with the InfoNCE objective (Gutmann and Hyvärinen, 2010; Oord et al., 2018), which is a lower bound on $I(x, \phi(x))$.

## 4 TASK-INDUCED REPRESENTATION LEARNING

An alternative class of representation learning approaches uses task information from prior tasks as a filter for which aspects of the environment are interesting, and which others do not need to be

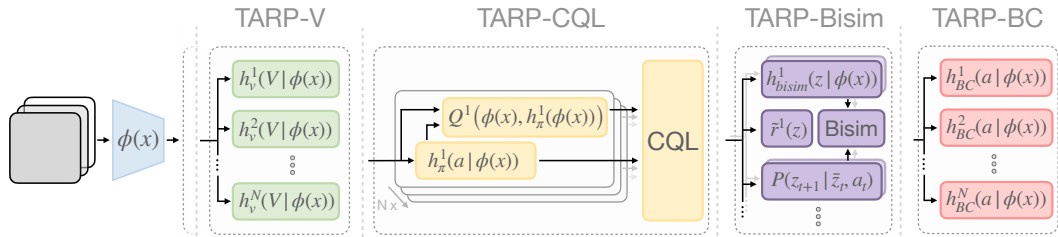

Figure 2: Instantiations of our task-induced representation learning framework. **Left to right**: Representation learning via multi-task value prediction (**TARP-V**), via multi-task offline RL (**TARP-CQL**), via bisimulation (**TARP-Bisim**) and via multi-task imitation learning (**right**, **TARP-BC**).

modeled. This addresses a fundamental problem of unsupervised representation learning approaches: by maximizing the mutual information between observations and representation they are trained to model every bit of information equally and thus can struggle in visually complex environments with lots of irrelevant details.

Formally, the state of an environment $\mathcal{S}$ can be divided into task-relevant and task-irrelevant or nuisance components $\mathcal{S} = \{\mathcal{S}_{\text{task}}^i, \mathcal{S}_n^i\}$. The superscript $i$ indicates that this division is *task-dependent*, since some components of the state space are relevant for a particular task $T_i$ but irrelevant for others. In visually complex environments, we can assume that $|\mathcal{S}_{\text{task}}| \ll |\mathcal{S}|$, i.e., that the task-relevant part of the state space is much smaller than all state information in the input observations (Zhaoping, 2006).

When choosing how much of this input information to model, end-to-end policy learning and unsupervised representation learning, form two ends of a spectrum. End-to-end policy learning only models $\mathcal{S}_{\text{task}}^i$ for its training task, which is efficient, but only allows transfer from task $i$ to task $j$ if $\mathcal{S}_{\text{task}}^j \subseteq \mathcal{S}_{\text{task}}^i$, i.e., if the task-relevant components of the target task are a subset of those of the training task. In contrast, unsupervised representation learning models the full $\mathcal{S} = \{\mathcal{S}_{\text{task}}, \mathcal{S}_n\}$, which allows for flexible transfer since $\mathcal{S}_{\text{task}}^j \subseteq \mathcal{S} \ \forall \ T_j \in \mathcal{T}$, but training can be inefficient since the learned representation contains many nuisance variables.

In this work, we formalize the family of task-induced representation learning (TARP) approaches, which aim to combine the best of both worlds. Similar to end-to-end policy learning, task-induced representation learning approaches use task information to learn compact, task-relevant representations. Yet, by combining the task information from a wide range of prior tasks they learn a representation that *combines* the task-relevant components of all tasks in the training dataset $\mathcal{D}$: $\mathcal{S}_{\text{TARP}} = \mathcal{S}_{\text{task}}^1 \cup \cdots \cup \mathcal{S}_{\text{task}}^N$. Thus, such a representation allows for transfer to a wide range of *unseen* tasks for which $\mathcal{S}_{\text{task}}^{\text{target}} \subseteq \mathcal{S}_{\text{TARP}}$. Yet, the representation filters a large part of the state space which is not relevant for *any* of the training tasks, allowing sample efficient learning on the target task.

### 4.1 APPROACHES FOR TASK-INDUCED REPRESENTATION LEARNING

We compare multiple approaches for task-induced representation learning which can leverage different forms of task supervision. We focus on reward annotations and task demonstrations; but, future work can extend the TARP framework to other forms of task supervision like language commands or human preferences. All instantiations of TARP train a shared encoder network $\phi(x)$ with separate task-supervision heads (see Figure 2).

**Value Prediction (TARP-V).** We can leverage reward annotations as task supervision, similar to Fu et al. (2021), by estimating the future discounted return of the data collection policy. Intuitively, a representation that allows estimation of the value of a state needs to include all task-relevant aspects of this state. We introduce separate value prediction heads $h_v^i$ for each task $i$ and train the representation $\phi(x)$ by minimizing the error in the predicted discounted return:

$$\mathcal{L}_{\text{TARP-V}} = \sum_{i=1}^{N} \mathbb{E}_{(x_t, r_t) \sim \mathcal{D}_i} \left( h_v^i(\phi(x_t)) - \sum_{t'=t}^{T} \gamma^{t'-t} r_{t'} \right)^2. \tag{1}$$

**Offline RL (TARP-CQL).** Alternatively, we can learn a representation by training a policy to directly maximize the discounted reward of the training tasks. Since we aim to use offline training

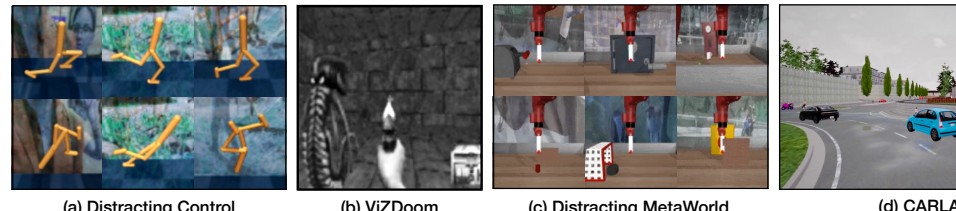

Figure 3: Environments with high visual complexity and substantial task-irrelevant detail for testing the learned representations. (a) **Distracting DMControl** (Stone et al., 2021) with randomly overlayed videos, (b) **ViZDoom** (Wydmuch et al., 2018) with diverse textures and enemy appearances, (c) **Distracting MetaWorld** (Yu et al., 2020) with randomly overlayed videos, and (d) **CARLA** (Dosovitskiy et al., 2017) with realistic outdoor driving scenes and weather simulation.

data only, we leverage recently proposed methods for offline RL (Levine et al., 2020; Kumar et al., 2020) and introduce separate policy heads $h_\pi^i$ and critics $Q^i$ for each task. Following Kumar et al. (2020), we train the policy to maximize the entropy-augmented future return:

$$\mathcal{L}_{\text{TARP-CQL}} = -\sum_{i=1}^{N} \mathbb{E}_{x \sim \mathcal{D}_i} \left( Q^i(x, h_\pi^i(\phi(x))) + \alpha \mathcal{H}\big(h_\pi^i(\phi(x))\big) \right). \tag{2}$$

Here $\mathcal{H}(\cdot)$ denotes the entropy of the policy's output distribution and $\alpha$ is a learned weighting factor (Kumar et al., 2020; Haarnoja et al., 2018b).

**Bisimulation (TARP-Bisim).** We use a bisimulation objective for reward-induced representation learning (Larsen and Skou, 1991; Ferns et al., 2011). It groups states based on their "behavioral similarity", measured as their expected future returns under arbitrary action sequences. Specifically, we build on the approach of Zhang et al. (2021) that leverages deep neural networks to estimate the bisimulation distance, and modify it for multi-task learning by adding per-task embedding heads $z^i = h_{\text{bisim}}^i\big(\phi(x)\big)$. The representation learning objective is:

$$\mathcal{L}_{\text{TARP-Bisim}} = \sum_{i=1}^{N} \mathbb{E}_{\substack{(x_j, a_j, r_j) \sim \mathcal{D}_i \\ (x_k, a_k, r_k)}} \left( |z_j^i - z_k^i| - |r_j - r_k| - \gamma \mathcal{W}_2\big(P(\cdot|\bar{z}_{j,t}^i, a_{j,t}), P(\cdot|\bar{z}_{k,t}^i, a_{k,t})\big) \right)^2 \tag{3}$$

Here $P(\cdot|z, a)$ is a learned latent transition model and $\mathcal{W}_2$ refers to the 2-Wasserstein metric which we can compute in closed form for Gaussian transition models. Following Zhang et al. (2021) we use a target encoder updated with the moving average of the encoder's weights for producing $\bar{z}$ and we add an auxiliary reward prediction objective with per-task reward predictors $\tilde{r}_{\text{bisim}}^i(z)$.

**Imitation Learning (TARP-BC).** We can also train task-induced representations from data without reward annotation by directly imitating the data collection policy, thus learning to represent all elements of the state space that were important for the policy's decision making. We choose behavioral cloning (BC, Pomerleau (1989)) for imitation learning since it is easily applicable especially in the offline setting. We introduce $N$ separate imitation policy heads $h_{\text{BC}}^i$ and minimize the negative log-likelihood of the training data's actions:

$$\mathcal{L}_{\text{TARP-BC}} = -\sum_{i=1}^{N} \mathbb{E}_{(x,a) \sim \mathcal{D}_i} \left( \log h_{\text{BC}}^i(a|\phi(x)) \right) \tag{4}$$

## 5 EXPERIMENTS

### 5.1 EXPERIMENTAL SETUP

We compare the performance of different representation learning approaches in four visually complex environments (see Figure 3). We will briefly describe each environment; for more details on environment setup and data collection, see appendix, Section A.

**Distracting DMControl.** We use the environment of Stone et al. (2021), in which the visual complexity of the standard DMControl "Walker" task (Tassa et al., 2018) is increased by overlaying randomly sampled natural videos from the DAVIS 2017 dataset (Pont-Tuset et al., 2017). We train on data collected from standing, forward and backward walking policies and test on a downstream running task. An efficient representation should focus on modeling the agent while ignoring irrelevant information from the background video.

**ViZDoom.** A simulator for the ego-shooter game Doom (Wydmuch et al., 2018). We use pre-trained models from Dosovitskiy and Koltun (2017) for data collection and vary their objectives to get diverse behaviors such as maximizing the number of collected medi-packs or minimizing the loss of health points. Learned representations should focus on important aspects such as the location of enemies or medi-packs, while ignoring irrelevant details such as the texture of walls or appearance features of medi-packs etc. We test on the full "battle" task from Dosovitskiy and Koltun (2017).

**Distracting MetaWorld.** Based on the MetaWorld multi-task robotic manipulation environment (Yu et al., 2020). We increase the visual complexity by overlaying the background with the same natural videos used in the distracting DMControl, but define a much larger set of 15 training and 6 testing tasks, which require manipulation of diverse objects with a Sawyer robot arm. All objects manipulated in the testing tasks are present in at least one of the training tasks. During evaluation, we report the average performance across the six testing tasks and normalize each task's performance with the score of the best-performing policy.

**Autonomous Driving.** We simulate realistic first-person driving scenarios using the CARLA simulator (Dosovitskiy et al., 2017). We collect training data from intersection crossings as well as right and left turns using pre-trained policies and test on a long-range point-to-point driving task that requires navigating an unseen part of the environment. For efficient learning, representations need to model relevant driving information such as position and velocity of other cars, while ignoring irrelevant details such as trees, shadows or the color and make of cars in the scene.

We compare the different instantiations of the task-induced representation learning framework from Section 4.1 to common unsupervised representation learning approaches:

- **Reconstruction**: We train a beta-**VAE** (Higgins et al., 2017) or a stochastic video prediction model ("**Pred-O**") on the training data and transfers the encoder.
- **Contrastive Learning**: We use the contrastive learning objective from Stooke et al. (2021) for pre-training ("**ATC**").

We also compare to an approach that combines video prediction with reward prediction for pre-training ("**Pred-R+O**"), similar to Hafner et al. (2019), thus combining elements form unsupervised and task-induced representation learning. Additionally, we report performance of a **policy transfer** baseline, which pre-trains a policy on $\mathcal{D}$ using BC and then finetunes the full policy on the downstream task as an alternative to representation transfer[2]. For environments that provide a low-dimensional state (DMControl, MetaWorld), we further report results for an **oracle** baseline.

After pre-training, we transfer the frozen encoder weights to the target task policy, which we train with soft actor-critic (SAC, Haarnoja et al. (2018a)) on continuous control tasks (distracting DMControl, distracting MetaWorld, CARLA), and with PPO (Schulman et al., 2017) on discrete action tasks (ViZDoom).[3] For more implementation details and hyperparameters, see Appendix D.

## 5.2 DOWNSTREAM TASK LEARNING EFFICIENCY

We report downstream task learning curves for task-induced and unsupervised representation learning methods as well as direct policy transfer in Figure 4. The low performance of the *policy transfer* baseline (purple) shows that in most tested environments the downstream task requires significantly different behaviors than those observed in the pre-training data, a scenario in which transferring

---

[2]We also tried pre-training policies on each of the individual task datasets $\mathcal{D}_1, \dots, \mathcal{D}_N$ but found the finetuning performance of the policies trained on the full dataset $\mathcal{D}$ to be superior.

[3]We report results with finetuning of the pre-trained encoder in appendix, Section C, and find no substantial difference to the setting with frozen encoder.

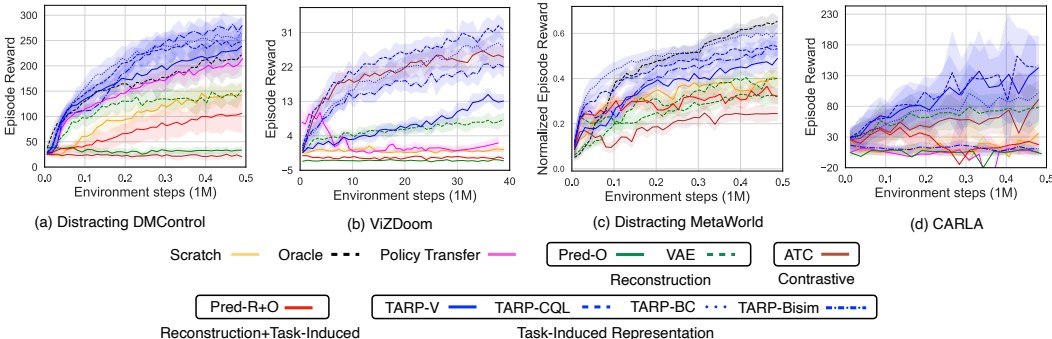

Figure 4: Performance of transferred representations on unseen target tasks. The task-induced representations (**blue**) lead to higher learning efficiency than the fully unsupervised representations or direct policy transfer and achieve comparable performance to the Oracle baseline. All results averaged across three seeds. See Figure 9 for per-task MetaWorld performances.

representations can be beneficial over directly transferring behaviors[4]. All reconstruction-based approaches (green) struggle with the complexity of the tested scenes, especially the method that attempts to predict the scene dynamics (*Pred-O*). We find that adding reward prediction to the objective improves performance (red, *Pred-O+R*). Yet, downstream learning is still slow, since the reconstruction objective leads to task-irrelevant information being modeled in the learned representation. The non-reconstructive contrastive approach *ATC* (brown) achieves stronger results, particularly in VizDoom which features less visual distractors, but its performance deteriorates substantially in environments with more task-irrelevant details, i.e. distracted DMControl, MetaWorld and CARLA.

Overall, we find that task-induced representations enable more sample efficient learning. Among the TARP instantiations, we find that TARP-V and TARP-Bisim representations can lead to lower transfer performance, since they rely on the expressiveness of the reward function: if future rewards can be predicted without modeling all task-relevant features, the representations will not capture them. In contrast, TARP-CQL and TARP-BC learn representations via policy learning, which can enable more efficient transfer to downstream policy learning problems. On the distracting DMControl task we find that TARP representations even outperform representations trained with direct supervision through a handcrafted oracle state representation, since they can learn to represent concepts like joint body parts of the walker during pre-training, while the oracle needs to learn these during downstream RL.

## 5.3 PROBING TASK-INDUCED REPRESENTATIONS

To better understand TARP's improved learning efficiency, we visualize what information is captured in the representation in Figure 5: we compare input saliency maps for representations learned with task-induced and unsupervised objectives. Saliency maps visualize the average gradient magnitude for each input pixel with respect to the output $\phi(x)$ and thus capture the contribution of each part of the input to the representation. We find that task-induced representations focus on the important aspects of the scene, such as the walker agent in distracting DMControl and other cars in CARLA. In contrast, the unsupervised approaches have high saliency values for scattered parts of the input and often represent task-irrelevant aspects such as changing background videos, buildings and trees, since they cannot differentiate task-relevant and irrelevant information.

For quantitative analysis, we train probing networks on top of the learned representations in distracting DMControl. We test whether (1) task-relevant information is modeled by predicting oracle joint states and (2) whether task-irrelevant information is ignored by classifying the ID of the used background video. The more irrelevant background information is captured in the representation, the better the probing network will be at classifying the video. The results show that probing networks trained with task-induced representations more accurately predict the task-relevant state information (Figure 6a) while successfully filtering information about the background video and thus obtaining lower background classification accuracy (Figure 6b).

---

[4]The policy transfer baseline achieves good performance in the distracting DMControl environment since it can reuse behaviors from the *walk-forward* training task for the target *run-forward* task.

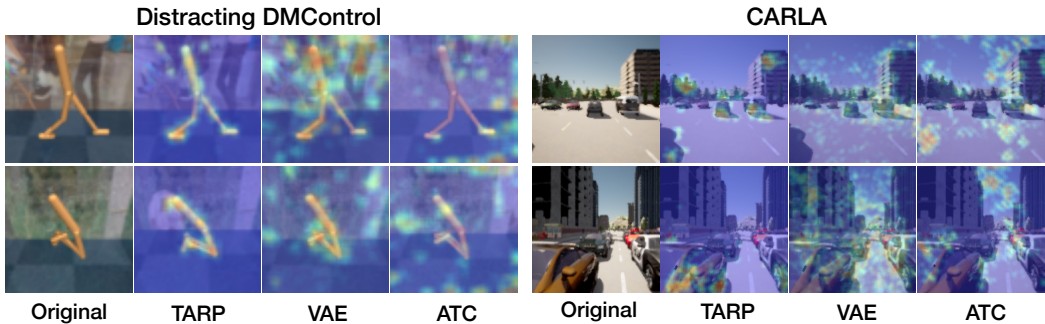

Figure 5: Visualization of the learned representations. **Left to right**: saliency maps for representations learned with task-induced representation learning (TARP-BC) and the highest-performing comparisons for reconstruction-based (VAE) and reconstruction-free (ATC) representation learning. **Left**: Distracting DMControl environment. **Right**: CARLA environment. Only task-induced representations can ignore distracting information and focus on the important aspects of the scene.

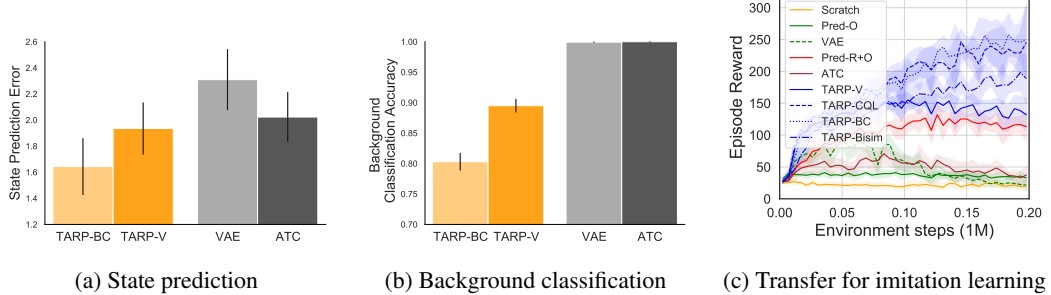

| (a) State prediction | (b) Background classification | (c) Transfer for imitation learning |

Figure 6: (a) Task-induced representations (TARP-BC/V) allow for more accurate prediction of the task-relevant joint states. Unsupervised approaches aim to model all information in the input – thus probing networks can still learn to predict state information, but struggle to achieve comparably low error rates. (b) Task-induced representations successfully filter task-irrelevant background information and thus cannot confidently classify the background video, while unsupervised approaches fail to filter the irrelevant information and thus achieve perfect classification scores. (c) Transfer performance for IL in distracting DMControl. Task-induced representations achieve superior sample efficiency.

## 5.4 TRANSFERRING REPRESENTATIONS FOR IMITATION LEARNING

We test whether the conclusions from the model-free RL experiments above equally apply when instead performing imitation learning (IL) on the downstream task. We train policies with the pre-trained representations from Section 5.2 by Soft-Q Imitation Learning (SQIL, Reddy et al. (2020)) on the distracting DMControl target running task. In Figure 6c we show that task-induced representations also improve downstream performance in visually complex environments for IL and allow for more efficient imitation, since they model only task-relevant information. Again, TARP-BC and TARP-CQL lead to the best learning efficiency. This shows that the benefit of modeling task-relevant aspects is not constrained to RL, but the *same* representations can be used to accelerate IL.

## 5.5 DATA ANALYSIS EXPERIMENTS

In the previous sections, we found that task-induced representations can improve learning efficiency over common unsupervised representation learning approaches in visually complex environments. These task-induced representation learning approaches are designed to leverage data from a variety of sources, such as prior training runs or human teleoperation. Thus, analyzing what characteristics of this training data lead to successful transfer is key to their practical use – our goal in this section is to derive a set of best practices when collecting datasets for task-induced representation learning.

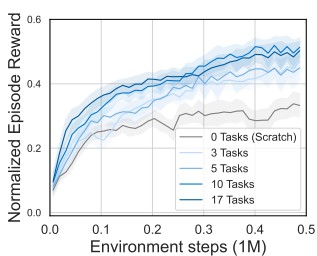

**Many tasks vs. lots of data per task.** When collecting large datasets there is a trade-off between collecting a lot of data for few tasks vs. collecting less data each for a larger set of tasks. To analyze the effects of this trade-off on TARP, we train TARP-BC on data from a varying number of pre-training tasks in distracting MetaWorld. When training on data from fewer tasks, we collect more data on each of these tasks, to ensure that the size of the training datasets is constant across all experiments. The results in Figure 7 show: **training on fewer data from a larger number of tasks is beneficial over training on lots of data from few tasks**. Intuitively, since downstream tasks can leverage the union of task-relevant components of all training tasks, having a diverse set of tasks is more important for transfer than having lots of data for few tasks.

Figure 7: Transfer performance vs. number of pre-training tasks. Collecting training data from a larger number of tasks is more important for transfer performance than having lots of data for few tasks.

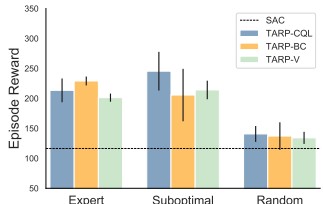

**More data vs. optimal data.** Another trade-off in data collection exists between the amount of data and its optimality: a method that can learn from sub-optimal trajectory data is able to leverage much larger and more diverse datasets. Since training on diverse data is important for successful transfer (see above) robustness to sub-optimal training data is an important feature of any representation learning approach. We test the robustness of task-induced representation learning approaches by training on sub-optimal trajectory data, collected from only partially trained and completely random policies, in distracting DMControl. The downstream RL performance comparison in Figure 8 shows that TARP approaches can learn strong representations from low-performance trajectory data and even when trained from random data, performance does not decrease compared

Figure 8: Robustness to pre-training data optimality. TARP approaches can learn good representations that allow for effective transfer even from sub-optimal data.

to a SAC baseline trained from scratch. Intuitively, task-induced representation learning does not pre-train a model on "how to act" but merely "what to pay attention to". We find that TARP-CQL's performance can even increase slightly with the suboptimal data, which we attribute to its increased state coverage. Thus we conclude that task-induced representation learning is robust to the optimality of the pre-training data and **collecting larger, diverse datasets is more important than collecting optimal data**.

**Multi-Source Task Supervision.** When collecting large datasets, it can be beneficial to pool data from multiple sources, e.g. data obtained from prior training runs or via human teleoperation. In Section 4 we introduced different instantiations of the TARP framework that are able to leverage different forms of task supervision. Here, we test whether we can train a single representation using multiple sources of task supervision simultaneously. In particular, we train "TARP-V+BC" models that assume reward annotations for only some of the tasks in the pre-training dataset and demonstration data for the remaining tasks (for more details on data collection, see Section B). We compare downstream learning efficiency on distracting DMControl and CARLA in Figure 10 We find that the combined-source model trained from the heterogeneous dataset achieves comparable or superior performance to all single-source models, showing that practitioners should **collect diverse datasets, even if they have heterogeneous sources of task-supervision**.

## 6 CONCLUSION

In this work, we investigate the effectiveness of representation learning approaches for transfer in visually complex scenes. We define the family of task-induced representation learning approaches, that leverage task-information from prior tasks to only model task-relevant aspects of a scene while ignoring distractors. We compare task-induced representations to common unsupervised representation learning approaches across four visually complex environments with substantial distractors and find that they lead to improved sample efficiency on downstream tasks compared to unsupervised approaches. Future work should investigate approaches for incorporating other sources of task information such as language commands to learn task-induced representation.

## 7 REPRODUCIBILITY STATEMENT

We provide a detailed description of all used environments, the procedures for offline dataset collection, and descriptions of the downstream evaluation tasks in appendix, Section A. Furthermore, in appendix, Section D we list all hyperparameters used for the pre-training phase (i.e. task-induced and unsupervised representation learning) as well as for training of the RL policies. We open-source our codebase with example commands to reproduce our results on the project website: clvrai.com/tarp.

## 8 ACKNOWLEDGEMENT

We thank our colleagues from the CLVR lab at USC for the valuable discussions that considerably helped to shape this work.

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

# A   ENVIRONMENTS

## A.1   DISTRACTING DMCONTROL

Following Stone et al. (2021), we increase the visual complexity of the DMControl "walker" task (Tassa et al., 2018) by overlaying randomly sampled videos from the DAVIS 2017 dataset (Pont-Tuset et al., 2017) in the background. In our experiments we use expert policies to collect offline datasets for the pre-training tasks of standing, forward walking, and backward walking respectively. To collect these datasets, we pre-train polices with SAC(Haarnoja et al., 2018a) in the state space and collect rollouts of the visual observation. We test our representation on the downstream task of "running".

**Rewards:** We use reward functions provided by DMControl "walker" task (Tassa et al., 2018). For the backward walking task, we invert the sign of the walking speed in the "forward" task defined in DMControl, so that the agent gets higher reward when it moves backwards instead of forwards.

## A.2   VIZDOOM

Our experiments use the "D3 battle"environment provided by Wydmuch et al. (2018) where the agent's objective is to defend against enemies while collecting medi-packs and ammunition. For collection of the pre-training task dataset, we use the pre-trained models provided by Dosovitskiy and Koltun (2017) and vary the reward weighting parameters (described below) to produce a diverse set of behaviors.

**Rewards:** A reward function is defined by a linear combination of three measurements (ammunition, health, and frags) with their corresponding coefficients.

$$R_{\text{ViZDoom}} = c_{ammo} \cdot \frac{x_t^{amm} - x_{t-1}^{amm}}{7.5} + c_{health} \cdot \frac{x_t^{health} - x_{t-1}^{health}}{30.0} + c_{frags} \cdot \frac{x_t^{frags} - x_{t-1}^{frags}}{1.0}. \tag{5}$$

where $x_t^{amm}$ is a measurement of ammunition, $x_t^{health}$ is the health of the agent, and $x_t^{frags}$ is the number of frags at timestep $t$. The set of coefficients for ammunition, health and frags are represented as $(c_{ammo}, c_{health}, c_{frags})$. We use the coefficients of $(0, 0, 1)$, $(0, 1, 0)$, and $(1, 1, -1)$ for collecting the pre-training data, and $(0.5, 0.5, 1.0)$ for the target task.

## A.3   DISTRACTING METAWORLD

We increase the visual complexity of MetaWorld environment (Yu et al., 2020) by overlaying randomly sampled videos from the DAVIS 2017 dataset (Pont-Tuset et al., 2017), similar to Stone et al. (2021). For data collection, we pre-train polices with SAC(Haarnoja et al., 2018a) using low-dimensional state information and collect rollouts of the visual observations. All objects manipulated in the downstream tasks are present in at least one of the training tasks. We define 15 tasks for data collection:

| | | |
|---|---|---|
| button press | button press topdown | button press topdown wall |
| plate slide | plate slide back | plate slide side |
| handle pull | handle pull side | handle press |
| door open | door lock | coffee button |
| coffee push | push | push back |

We evaluated transfer performance on 6 downstream tasks:

| | | |
|---|---|---|
| button press wall | plate slide back side | handle press side |
| door unlock | coffee pull | push wall |

We report the average performance across the six downstream tasks and normalize each task's performance with the episode reward of the best-performing policy. Furthermore, we show the performance of each downstream task in Figure 9.

**Rewards:** We use reward functions provided by Metaworld (Yu et al., 2020).

## A.4 CARLA

We use the map of "Town05" from the CARLA environment (Dosovitskiy et al., 2017) for our experiments. At the beginning of each episode, we randomly spawn 300 vehicles and 200 pedestrians. The initial location of the agent is randomly sampled from a task set containing multiple start and goal locations. We collect pre-training datasets for the tasks of intersection crossing, taking a right turn, and taking a left turn using pre-trained policies. Then, we test on a long-range curvy road point-to-point driving task that requires navigating an unseen part of the environment. We pre-train the policies with SAC (Haarnoja et al., 2018a) using segmentation masks of the environment as the input, and then use the learnt policies to collect rollouts of visual observations for the datasets.

**Rewards:** We modify a reward function used in Vergara (2019) for all of the tasks. The reward function consists of terms for speed, centering on a road, angle of the agent, and collision.

$$
\begin{aligned}
R_{\text{speed}} &= \frac{v}{v_{min}} \cdot \mathbb{1}_{v \leq v_{min}} + (1.0 - \frac{v - v_{target}}{v_{max} - v_{target}}) \cdot \mathbb{1}_{v \geq v_{max}} + 1.0 \cdot \mathbb{1}_{v_{min} \leq v \leq v_{max}}. \\
R_{\text{centering}} &= \max(1.0 - \frac{d_{center}}{d_{max}}, 0). \\
R_{\text{angle}} &= \max(1.0 - |\frac{r}{(r_{max} \cdot \frac{\pi}{180})}|, 0). \\
R_{CARLA} &= R_{\text{speed}} + R_{\text{centering}} + R_{\text{angle}} - 10^{-4} \cdot collision\_intensity.
\end{aligned}
\tag{6}
$$

where $v$ is velocity of the agent, $d_{center}$ is distance between the center of the road and the agent, $r$ is angle of the agent. We use constant values of $v_{min} = 15.0$, $v_{max} = 30.0$, $v_{target} = 25.0$, $d_{max} = 3$, and $r_{max} = 20$.

## B   DATA COLLECTION FOR MULTI-SOURCE TASK SUPERVISION

We detail the composition of the pre-training dataset for the experiment in which we train TARP from heterogeneous data sources. For the distracting DMControl environment, the dataset is composed of demonstrations for the "forward walking" and "backward walking" tasks, and a dataset with reward annotations for the "stand" task. For the CARLA environment, the task-induced representations are trained on a dataset composed of demonstrations for the "intersection crossing" task, and data annotated with rewards for the "right turn" and "left turn" tasks. For both environments, the datasets are collected with the procedures described in appendix, Section A.

## C   FINETUNING LEARNED REPRESENTATIONS

In our experimental evaluation in Section 5 we held the parameters of the pre-trained encoder fixed to cleanly evaluate the quality of the pre-trained representation. However, in practice pre-trained representations are often finetuned on the target task using target task rewards. Prior work on representation learning found mixed results when finetuning the pre-trained representations (Yang and Nachum, 2021). Thus, in this section we experimentally compare the different representation learning approaches on the Distracting DMControl, ViZDoom, and CARLA environment *while finetuning the learned representations on the target task*. As illustrated in Figure 11(a), we find that task-induced representations show improved sample efficiency and better performance even in the fine-tuning setting, analogous to the results in Section 5.2.

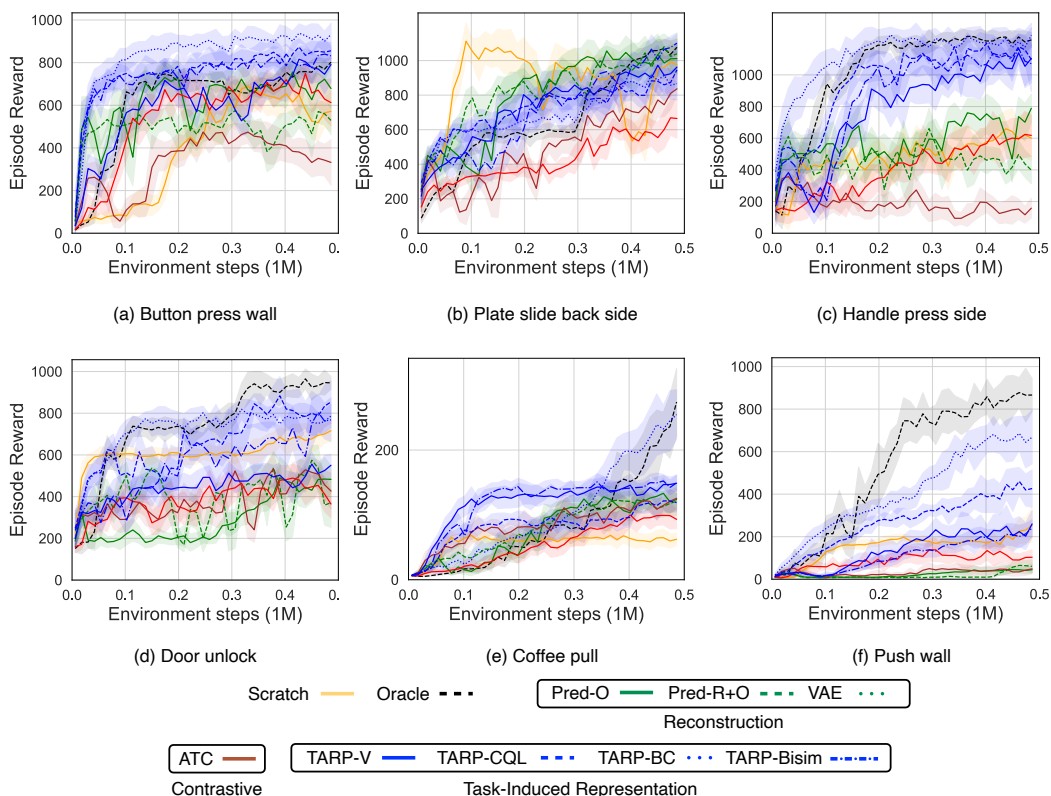

Figure 9: Transfer performance for each downstream task in the distracting MetaWorld environment. The task-induced representations generally show better performance than unsupervised representations and achieve comparable performance to the oracle baseline.

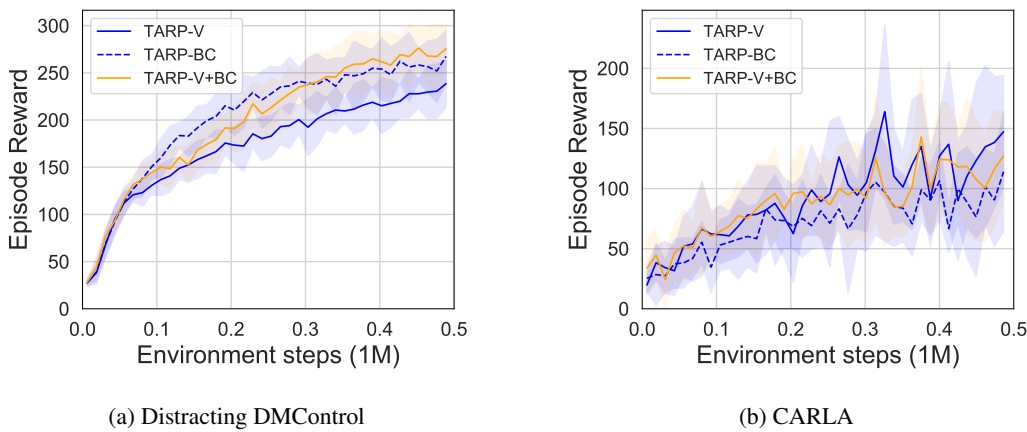

Figure 10: Transfer performance for single source and multiple source task-induced representation. The task-induced representation with multiple supervision sources performs slightly better or as good as the single source representation. This shows that task-induced representations can be effectively trained from datasets with heterogeneous forms of task supervision.

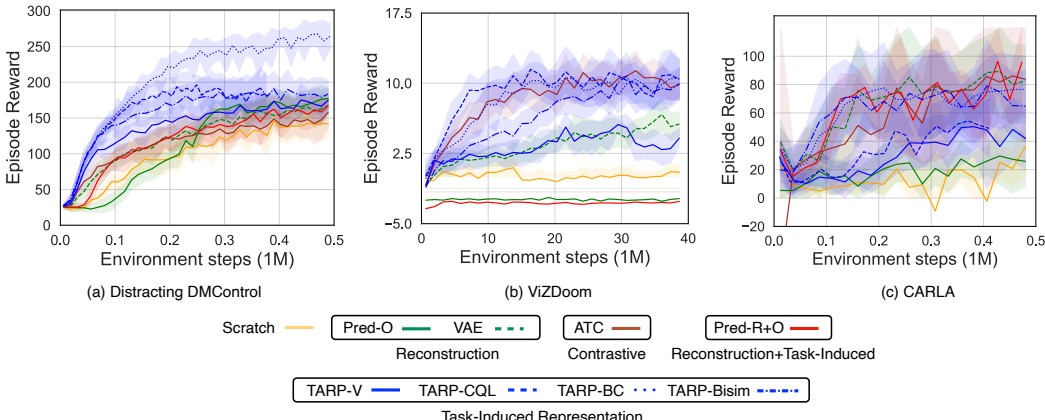

Figure 11: Performance of transferred representations with finetuning on unseen target tasks on distracting DMControl, ViZDoom, and CARLA environments. The task-induced representations (**blue**) show better sample efficiency and performance compared to the representations learned with unsupervised objectives.

# D HYPERPARAMETERS

## D.1 HYPERPARAMETERS FOR RL

Table 1: Common SAC hyperparameter

| Parameter | Value |
|---|---|
| Stacked frames | 3 |
| Optimizer | Adam |
| Learning rate | 3e-4 |
| Discount factor ($\gamma$) | 0.99 |
| Latent dimension | 256 |
| Convolution filters | $[8, 16, 32, 64]$ |
| Convolution strides | $[2, 2, 2, 2]$ |
| Convolution filter size | 3 |
| Hidden Units (MLP) | $[1024]$ |
| Nonlinearity | ReLU |
| Target smoothing coefficient ($\tau$) | 0.005 |
| Target entropy | $-\dim(\mathcal{A})$ |

Table 2: Distracting DMControl and MetaWorld SAC hyperparameter

| Parameter | Value |
|---|---|
| Observation Rendering | (64, 64), RGB |
| Initial steps | $5 \times 10^3$ |
| Action repeat | 2 |
| Replay buffer size | $10^5$ |
| Minibatch size | 256 |
| Target update interval | 1 |
| Actor update interval | 1 |
| Initial temperature | 1 |

Table 3: CARLA SAC hyperparameter

| Parameter | Value |
|---|---|
| Observation Rendering | (128, 128), RGB |
| Initial steps | $3 \times 10^3$ |
| Action repeat | 1 |
| Replay buffer size | $10^5$ |
| Minibatch size | 128 |
| Target update interval | 2 |
| Actor update interval | 2 |
| Initial temperature | 0.1 |

Table 4: ViZDoom PPO hyperparameter

| Parameter | Value |
|---|---|
| Observation rendering | (64, 64) Grey |
| Stacked frames | 4 |
| Action repeat | 1 |
| Optimizer | Adam |
| Learning rate | 3e-4 |
| PPO epoch | 10 |
| Buffer size | 2048 |
| Convolution filters | $[8, 16, 32, 64]$ |
| Convolution filter sizes | $[2, 2, 2, 2]$ |
| Hidden units (MLP) | $[256]$ |
| Generalized advantage estimation $\lambda$ | 0.95 |
| Entropy bonus coefficient | 4e-3 |
| Discount factor ($\gamma$) | 0.99 |
| Minibatch size | 256 |
| Nonlinearity | ReLU |

Table 5: Hyperparameters for CQL

| Environment | Trade-off factor $\alpha$ for Q-values | Number of action samples |
|---|---|---|
| Distracting DMControl | 3. | 1 |
| ViZDoom | 1. | 1 |
| CARLA | 3. | 1 |

## D.2 HYPERPARAMETERS FOR PRE-TRAINING

In TARP-BC, we use recurrent neural networks to predict a sequence of actions over prediction horizon $T$ to learn task-induced representations only in CARLA (see Table 7).

Table 6: Common model parameters

| Parameter | Value |
|---|---|
| Batch size | 128 |
| Hidden units (MLP) | $[256]$ |
| Learning rate | 1e-4 |

Table 7: Hyperparameters for TARP-BC

| Environment | LSTM hidden units | Prediction horizon |
|---|---|---|
| Distracting DMControl | — | - |
| ViZDoom | — | - |
| Distracting MetaWorld | — | - |
| CARLA | 512 | 8 |

Table 8: Hyperparameters for TARP-V

| Environment | Discount rate |
|---|---|
| All environments | 0.4 |

Table 9: Hyperparameters for TARP-Bisim

| Environment | Reward predictive loss weight | Bisimulation loss weight $T$ |
|---|---|---|
| Distracting DMControl | 1. | 0.1 |
| ViZDoom | 1. | 10. |
| Distracting MetaWorld | 1. | 0.1 |
| CARLA | 1. | 0.01 |

Table 10: Hyperparameters for VAE

| Environment | $\beta$ constraint |
|---|---|
| Distracting DMControl | 100. |
| ViZDoom | 100. |
| Distracting MetaWorld | 100. |
| CARLA | 10. |

Table 11: Hyperparameters for Pred-S and Pred-R+S

| Environment | $\beta$ constraint | Prediction horizon $T$ |
|---|---|---|
| Distracting DMControl | 50. | 6 |
| ViZDoom | 10. | 6 |
| Distracting MetaWorld | 50. | 6 |
| CARLA | 5. | 8 |

Table 12: Hyperparameters for ATC

| Environment | Random shift probability | Temporal shift |
|---|---|---|
| All environments | 1. | 3 |

