# OpenReview forum: "Task-Induced Representation Learning"
_ICLR.cc/2022/Conference — ICLR 2022 Poster_

### Official Review · Reviewer_k7xA · 2021-10-28

**Correctness:** 3
**Technical Novelty And Significance:** 1
**Empirical Novelty And Significance:** 2
**Recommendation:** 5
**Confidence:** 4

**Main Review:**

Advantages:

- I found the analysis combining saliency maps and state predictions vs background video classification interesting. It was nice to see the authors attempting to gain intuition by combining different techniques.
- The authors attempt to cover a number of interesting questions, from what should we predict, to how homogeneous or optimal the dataset should be. I found this interesting and informative.

Disadvantages:

- The authors only chose a single training-testing transfer pair of tasks from each environment type. Given how different the rankings of the performance of the methods are between each environment, I am curious whether there’s a high variance between different training-testing transfer pairs from the same environment. Though the results and narrative make sense, I am wondering whether the conclusions would hold in additional training-testing pairs within these three environments.
- I’m not clear what Section 5.4 and figure 6c show. What are the exact training-testing settings here? How do they differ from the experiments whose results are shown in figure 4?
- Though the authors attempt to answer many questions, they only provide analysis results on some of the methods. For example, how do saliency maps and state vs background predictions of every other method compare? Wasn’t it possible to undertake a similar analysis for non-TARP methods too?
- I recommend rewriting the contributions paragraph. Points 1 and 3 seem to be very similar and I don’t think the grouping provided by point 2 is necessary. Recognising that task relevant information is important for representation learning is a worthy contribution, but in my humble opinion “formalize task induced representation learning (TARP) as an alternative family of representation learning approaches that leverage task information from prior tasks“ is not really a separate contribution.


**Summary Of The Paper:**

The paper attempts to answer the question of how to pre-train encoders in order to get task-relevant representations that are useful for a downstream task in each of three different environments (ViZDoom,  Distracting DMContro and Autonomous Driving). Three main approaches are compared (Behavioral Cloning, Value prediction only or another commonly used offline RL loss), along with a number of additional baselines (VAEs, predicting combinations of observations and rewards, contrastive methods and others), in a setting where an encoder is pre-trained offline on data coming from a certain task and its weights are frozen and used to provide representations to a policy head trained to solve an unseen task, that’s held out in the original dataset. The authors also provide an analysis using saliency maps to explore which parts of the input image are important for decreasing the loss of each method (task relevant vs background) and also show the performance of the representation on a task relevant and a task irrelevant task. Finally, the authors explore how the quality of the data (coming from many tasks vs coming from a single task) during the representation learning phase, and whether they come from an optimal, random or suboptimal policy, affects downstream performance. The main contribution of the paper, in my opinion, is this investigation of the different methods. In particular the comparison of performance in each of the three environments, the saliency analysis and predictability of state vs background information and finally the exploration of the effect of data on performance.

**Summary Of The Review:**

In summary, I think the paper is interesting, but importantly, I did not find many of the findings surprising enough to recommend for acceptance. There are additionally two main issues with the paper in its current form. First, the contributions section in the introduction should, in my opinion, be re-written (see disadvantages section above). Second, experimentally there are results on only a single training-testing transfer pair for each of three environments. I would expect a more thorough investigation of multiple of these pairs to assess the robustness of methods that make use of task information.

---

> ### Author Response · Authors · 2021-11-23
> **Rebuttal — k7xA**
>
> We thank the reviewer for valuable feedback. We first want to clarify the scope of our submission. In this paper, our goal is to investigate the role of task information in representation learning, specifically in visually complex environments with distractors, rather than proposing fundamentally new approaches. We also provide a coherent framework that allows to capture many prior task-induced representation learning methods and allows to easily compare them. Following the reviewer’s feedback we have adjusted the contributions paragraph of the introduction to more clearly reflect this focus.
>
>
> **Does the conclusions hold in different training-test task pairs?**
>
> In general, the choices of pre-training vs. target tasks are interchangeable as long as the union of task-relevant features of the pre-training tasks covers the task-relevant features of the target task, or formally Stasktarget Stask1...StaskN   (see Section 4). Yet, in our evaluations we go beyond this and choose the target task such that it is more complex for the agent to learn than any of the pre-training tasks to evaluate the different approaches in a challenging setting. Concretely, we require the agent to run faster than any of the pre-training tasks in Distracting DMControl, to complete the full battle task after being trained on simpler subtasks in Vizdoom and to navigate longer episodes than any of the pre-training tasks in CARLA.
>
> However, to experimentally validate that we can interchange pre-training vs target tasks we include a new experiment in Appendix D where we change the pre-training vs target task sets in the Distracting DMControl environment: we pre-train the agent using the “run”, “stand” and “backwards” tasks and perform downstream training on the “walk” task. The quantitative results show that the same conclusions apply in this scenario: task-induced representations lead to more efficient downstream training than unsupervised alternatives.
>
>
> **What do Section 5.4 and figure 6c show? Any difference from Figure 4?**
>
> In section 5.4 and Figure 6c, we use exactly the same datasets, prior tasks, and downstream task as the experiments in Section 5.2 (Figure 4). However, In section 5.4, we assume that we have access to expert actions in the downstream task and apply imitation learning to show that learned representations are general and the conclusions drawn apply for different downstream learning problems (i.e. RL vs LfD).
>
> **Wasn’t it possible to undertake a similar analysis for non-TARP methods too**
>
> In our analysis experiments in Sections 5.3, we compare a representative set of approaches from each of the major classes of representation learning methods, covering task-induced representation learning methods (TARP-BC/V) as well as unsupervised-reconstructive (VAE) and unsupervised-contrastive (ATC) approaches. This allows us to investigate *why* task-induced representations lead to more sample efficient downstream learning in visually complex environments (see Figure 4). We focus the dataset analysis in Section 5.5 on task-induced representation learning approaches since, in the tested visually complex environments, we found them to enable the most efficient learning of target tasks.
>
> ---------------
>
> We again appreciate you for your valuable feedback, please let us know if this addresses your concerns and please consider updating your score accordingly!

---

> > ### Comment · Reviewer_k7xA · 2021-11-25
> > **Thank you for your response.**
> >
> > I would like to thank the authors for their response to my questions.
> >
> > As a follow up to my first question "Does the conclusions hold in different training-test task pairs?": I am curious on your thoughts on the following possible experiment: What if one were to test how well your findings transfer to the Atari domain, by training the representation on data from one level and then assess how useful these representations are for learning in another level. Do you think they would observe significant gains in performance compared to other representation learning methods? Even though the different levels do not necessarily completely share the same task features, I believe they share them to a large degree.
> >
> > I would be interested in your thoughts.

---

> > > ### Author Response · Authors · 2021-11-26
> > > **Re: Thank you for your response.**
> > >
> > > Thank you for this question! Task-induced representation learning approaches perform best when the following two conditions are met: (1) there is experience from multiple prior tasks available from which we can learn the representation, (2) environments are visually complex with lots of task-irrelevant information. Point (1) ensures that the learned representation captures a range of task-relevant features, thus likely covering the task-relevant aspects of the downstream task. Moreover, if the environment was not visually complex or featured distractors, task-induced representations would still work well, but the benefit over unsupervised representation learning methods that model everything in the scene would not be significant.
> > >
> > > In the case of Atari, most environments are designed for one particular task (e.g. hitting blocks in Breakout or destroying spaceships in Space Invader). Yet, as you suggested, some games like Montezuma’s revenge have multiple levels and one could use experience from different levels for learning task-induced representations. When trained across a diverse set of levels, we would expect that this would lead to representations that can transfer well to new levels. However, the visual complexity and number of distractors in Atari games is comparably low, possibly due to graphics capability limitations of game consoles in the 80s. Thus, most of the elements in the agent’s observation are task-relevant and we would therefore expect that the benefit of task-induced representations over unsupervised representations could be limited in these games.
> > >
> > > However, there are benchmarks that use more recent video games, like the Gym Retro [1] benchmark, which uses the “Sonic the Hedgehog'' game and features more advanced graphics that better approximate the complexity of the real world. There, we would expect to see a larger benefit of task-induced representations over unsupervised alternatives since they can learn to ignore the task-irrelevant details using experience from prior levels of the game.
> > >
> > > We hope that this answers your question. Please let us know if you have any further questions!
> > >
> > > [1] Nichol et al., “Gotta Learn Fast: A New Benchmark for Generalization in RL”

---

> > > > ### Comment · Reviewer_k7xA · 2021-11-29
> > > > **Thank you**
> > > >
> > > > I would like to thank the reviewers for the response.

---

### Official Review · Reviewer_TaF6 · 2021-11-02

**Correctness:** 3
**Technical Novelty And Significance:** 2
**Empirical Novelty And Significance:** 2
**Recommendation:** 6
**Confidence:** 4

**Main Review:**

Strengths:
1.	The paper presents an interesting discussion on task relevant supervision vs. purely unsupervised techniques. The empirical results are mostly in support of the proposed argument and contain diverse domains of tasks that the technique was tested on.
2.	The paper follows up the main results with with an additional analysis and discussion of why TARP techniques perform better than the baselines, specifically, the role data plays in this performance boost. This analysis adds value by showing a closer look at the features and the kind of data that would result in better performance; and is valuable for the representation learning community in general.

Weaknesses:
1.	Firstly, the crux of the paper deals with a comparison between using task-based supervision to learn representations, and unsupervised representation learning techniques. While the argument that task supervision is more valuable compared to unsupervised learning is interesting, there is an obvious difference in the sense that task supervision directly allows for extracting task-relevant features in a representation. When unsupervised learning techniques do not account for any relevance, or there are no particular constraints as in techniques like beta VAE, it is obvious that those representations would not fare as well as a task-induced one in the absence of any additional supervision, constraints, hand crafted loss functions etc. In that way, it is not exactly an apples-to-apples comparison. This is made evident from the saliency map comparison, in the absence of any finetuning / pre-defined constraints, there is no way for an unsupervised learning method to know which features are important for an arbitrary task.

This is why I feel that the discussion of novelty and the comparison with existing techniques is somewhat insufficient. For instance, it would have been more interesting and valuable to compare the proposed technique with algorithms such as DREAMER [1,2].  DREAMER also contains similar subgoals as TARP such as a value learning network. Although it is true that TARP proposes training value prediction or reward maximization on a ‘set’ of tasks as opposed to a single task, how is that better than, for instance, applying DREAMER itself to multiple tasks and expecting it to build a generalizable representation? Similarly, another relevant algorithm would be TIA (Task induced abstractions) [3] that proposes targeted reconstructions to solve issues with generalization as well as distractors and outperforms DREAMER. Comparing with other techniques in similar classes would help gauge the novelty and efficacy of the current method and localize it properly within the current literature.

2.	Secondly, there are some parts of the evaluation I am not clear about. What is the main intuition for why the baseline representations are unable to focus on task relevance during the policy learning process? During the pretraining phase, an unsupervised model likely encodes both task-relevant and task irrelevant features in each scene – which gives rise to two possibilities. Either the model is failing to encode features in a disentangled way, which ruins performance during finetuning because it is not able to isolate task-relevant features from the others. If that is not the case, then it should be possible to isolate which of the encoded features are useful during finetuning for a task. If the baseline models can do the latter, then they should be applicable for diverse tasks similar to the way TARP is claimed to be. Hence, I think it would be very valuable to have some experiment that includes finetuning the representations as well while learning the policy. Speaking just conceptually, if we consider only a single task, it is not clear why TARP based learning is fundamentally different from finetuning a representation learnt in an unsupervised way for a given task, assuming that the representation is a well structured one.


3.	I think the paper would also benefit from a discussion of how the proposed techniques could compare against the current increasing push towards unsupervised pretraining. Unsupervised pretraining through large “foundation models”, as well as reward-free pretraining are gaining traction in the world of representation learning and reinforcement learning, where specific tasks or rewards are not considered, and in cases were found to benefit downstream reinforcement learning or imitation learning [4]. In the current paper, the authors mention through empirical findings that it is more important to obtain (although less) data from a large number of tasks, as opposed to lots of data from few tasks. But this also means a lot of effort and handcrafting in setting up the tasks, the associated rewards and so on.

Minor comments/clarification questions:

1. What is the downstream task corresponding to Figure 7?
2. It is seen that the contrastive learning approach (ATC) performs really badly on distracting DMControl. How much of that is due to a lack of data augmentation? Would a contrastive learning approach perform significantly better if it was trained on a sufficiently large dataset with multiple distractors?
3. In Figure 8, is there a significant difference in the size of datasets between expert and suboptimal?

[1] Hafner, D., Lillicrap, T., Ba, J., & Norouzi, M. (2019). Dream to control: Learning behaviors by latent imagination. arXiv preprint arXiv:1912.01603.
[2] Hafner, D., Lillicrap, T., Norouzi, M., & Ba, J. (2020). Mastering atari with discrete world models. arXiv preprint arXiv:2010.02193.
[3] Fu, X., Yang, G., Agrawal, P., & Jaakkola, T. (2021, July). Learning task informed abstractions. In International Conference on Machine Learning (pp. 3480-3491). PMLR.
[4] Yang, M., & Nachum, O. (2021). Representation matters: Offline pretraining for sequential decision making. arXiv preprint arXiv:2102.05815.

**Summary Of The Paper:**

This paper discusses the problem of learning meaningful representations that can efficiently focus only on the features that are relevant for downstream tasks. As opposed to unsupervised representation learning approaches that do not differentiate between task-specific and other information in a given dataset, the paper proposes a framework named “Task Induced Representation Learning” which leverages task information to guide the representation learning. Through this framework, the paper suggests three possible ways to induce task-relevance in representations: by predicting the value of a state, maximizing the discounted reward for a group of tasks, or by imitating an expert policy again for a group of tasks. Through a set of experiments in three domains: distracting DMControl, VizDoom and CARLA, the authors show higher task learning efficiency using the proposed framework as compared to standard approaches such as reconstruction, contrastive learning, state-reward prediction etc.

**Summary Of The Review:**

This paper proposes a framework called Task-induced representation learning (TARP), with the claim that having task supervision during representation learning is a lot more beneficial than unsupervised, task-free representation learning. Moreover, TARP is able to use supervision from a set of multiple, perhaps diverse tasks so that the representations can benefit from different kinds of guidance and thereby encode features that are useful for downstream tasks. Through a set of experiments on DMC, VizDoom, CARLA etc. the authors show that TARP representations outperform usual unsupervised learning approaches. While the paper is well written and the discussion is interesting, it is well known that task induced supervision will induce more relevance compared to unconstrained, unsupervised learning, and I feel the paper is not strong enough because it does not compare to more relevant baselines such as DREAMER, Task Informed Abstractions.

---

> ### Author Response · Authors · 2021-11-23
> **Rebuttal — TaF6 (2/2)**
>
> **Comparison with unsupervised representation learning techniques?**
>
> Thank you for raising this important point! We agree that unsupervised learning methods are appealing for representation learning due to their scalability. We thus include comparisons to multiple unsupervised representation learning approaches, including contrastive learning [1] which has recently seen impressive applications in computer vision and RL [2, 3, 4]. However, while these approaches have previously only been investigated in clean lab settings in the context of RL, we find in our evaluation in visually more complex environments that aforementioned unsupervised methods can struggle to enable efficient downstream learning. We hypothesize that this is because they cannot distinguish which information is important to model and support this intuition with experimental analysis (see Sections 5.3).
>
> In contrast, task-induced representation learning methods provide an approach for learning what information is relevant in a scene and our experimental evaluation finds that this can lead to superior learning efficiency on new tasks when tested in visually complex environments.
> We agree with the reviewer that these methods make additional assumptions over unsupervised representation learning techniques, e.g. that the pre-training data is annotated with rewards. However, this is an assumption commonly made for example within the offline RL community [5, 6, 7], where we can e.g. assume that the data was collected during previously conducted experiments while learning other tasks and thus already comes with rewards (see L113-115). Indeed, task-induced representation learning approaches make less assumptions than offline RL methods since they do not require annotations with *target task* rewards but can instead directly use the existing rewards collected with the data.
>
> **What is the downstream task corresponding to Figure 7?**
>
> We tested on a “running” task in distracting DMControl for the experiments in Figure 7. We updated the paper to explicitly mention this. Thank you for pointing this out.
>
> **Why does ATC perform badly on distracting DMControl? Could more data help?**
>
> Since contrastive learning is an unsupervised representation learning approach it cannot distinguish between task-relevant and irrelevant information and thus struggles to model visually complex scenes like the distracting DMControl suite [8] we test in, which includes a large number of distracting background videos. We believe that this environment is particularly hard for contrastive approaches since any of this distracting information can be used to distinguish images from different sequences, reducing the incentive to model task-relevant aspects. In contrast, alternative unsupervised approaches that e.g. leverage reconstruction objectives for learning representations could, somewhat counter-intuitively, benefit from the large randomness of the background since it is too hard to model while the task-relevant part of the scene, the agent, is the only aspect that’s present in all scenes and thus more likely to be modeled. Yet, the overall scene complexity in this dataset still makes reconstruction-based approaches struggle to enable efficient downstream learning, as can be seen in our evaluation in Figure 4.
>
> We do not believe that the dataset size is a limiting factor for the contrastive learning approaches here: we use a pre-training dataset of 900k images in Distracting DMControl, which sufficiently captures the diversity in agent poses and backgrounds.
>
> **In Figure 8, is there a difference in the datasets size between expert and suboptimal?**
>
> We used the same size of datasets for all settings: expert, suboptimal, and random.
>
>
> -----------------------------------
> We again thank you for your valuable feedback, please let us know if this addresses your concerns and please consider updating your score accordingly!
>
> [1] Stooke et al. “Decoupling representation learning from reinforcement learning’ \
> [2] He et al. “Momentum Contrast for Unsupervised Visual Representation Learning” \
> [3] Chen et al. “A Simple Framework for Contrastive Learning of Visual Representations” \
> [4] Laskin et al. “CURL: Contrastive unsupervised representations for reinforcement learning ” \
> [5] Levine et al. “Offline Reinforcement Learning: Tutorial, Review, and Perspectives on Open Problems” \
> [6] Gulcehre et al. “RL Unplugged: A Suite of Benchmarks for Offline Reinforcement Learning” \
> [7] Sinha et al. “S4RL: Surprisingly Simple Self-Supervision for Offline Reinforcement Learning in Robotics”  \
> [8] Stone et al. “The Distracting Control Suite -- A Challenging Benchmark for Reinforcement Learning from Pixels”

---

> > ### Comment · Reviewer_TaF6 · 2021-11-27
> > **Reviewer Response**
> >
> > Thank you for your detailed response addressing my questions and concerns. I particularly appreciate the additional experiments that compared the proposed approach to TIA, as well as the performance of finetuning vs. training from scratch. It is also interesting to note the comment on how unsupervised learning approaches might learn task-relevant parts, perhaps an interesting direction for future work is to consider more unsupervised pretraining approaches that place emphasis on disentanglement of features.
> >
> > I have two clarification questions, I would be interested in your thoughts about those:
> >
> > 1. I find it odd that TIA, which seemingly demonstrates excellent foreground-background disentanglement (Figure 10), exhibits poorer downstream training performance than the beta VAE. Any thoughts on why this might be? Also, the comment that “TIA might be modeling task-irrelevant agent appearance” is not very clear, as it seems to be at odds with the statement “We find that task-induced representations can focus only on the important aspects of the scene, such as the walker agent in distracting DMControl”. Isn’t that what TIA is doing too, and wouldn’t the TIA saliency map be similar to, say, TARP-BC?
> >
> >
> > 2. In the experiment related to finetuning, are the TARP approaches (which are originally trained on a collection of tasks) being finetuned for the specific task as well? In an ideal setting, I would expect finetuning TARP to be less necessary compared to the other non-task-induced approaches, as TARP is already likely learning task-specific features (but I do agree that finetuning could help focus on a specific task further). Could the authors perhaps comment on the performance of non-finetuned TARP vs. finetuned non-task-induced representations?

---

> > > ### Author Response · Authors · 2021-11-28
> > > **Re: Reviewer Response (2/2)**
> > >
> > > **Finetuning experiments**: In the finetuning experiments we finetune all approaches, including the different TARP versions, using rewards from the target downstream task by propagating gradients from policy and critic into their respective pre-trained encoders (see Appendix C  for a detailed description). Regarding the need for finetuning your intuition is correct: since TARP representations are already focussing on task-relevant features they should require less finetuning on the target task, but as you mentioned the finetuning can still help since not all features learned in TARP representations are relevant to the particular downstream task at hand.
> > >
> > > Comparing RL with pre-trained (non-finetuned) TARP representations from Figure 4 to RL with finetuned unsupervised representations in Figure 11 we find that the TARP representations lead to better downstream performance. In practice, the unsupervised approaches start RL training with a comparatively worse representation that mixes task-relevant and irrelevant features and learning to filter out the irrelevant parts via finetuning of the encoder requires many training iterations. So while one could expect the finetuned representations to eventually lead to better downstream performance than the frozen TARP representation (since the former effectively has more trainable parameters), we did not see it outperform TARP within the horizon of our evaluation. In practice it is also possible that the finetuned representation never reaches superior performance due to learning dynamics in the RL training loop -- e.g. it is possible that the policy converges to a suboptimal local optimum before the optimal representation is reached via finetuning. Since such effects are tightly tied to e.g. the environment’s reward density it is likely not possible to draw general conclusions about a comparison of frozen TARP vs finetuned unsupervised representations, but we hope that this addresses the reviewer’s question!

---

> > > > ### Comment · Reviewer_TaF6 · 2021-11-29
> > > > **Reviewer response to comments**
> > > >
> > > > Thank you again for the detailed response. I understand the TIA vs. TARP comparison better now, where "visual appearance of the agent" is less important/beneficial than actual task-relevant state information, which might be resulting in worse performance for TIA. I am still a bit unclear about TIA being worse than beta-VAE which has completely unguided reconstructions, which is something that can perhaps be investigated through more experiments in the future. I would recommend adding these potential explanations of TIA's performance to the paper to help readers understand why task-inspired _reconstruction_ might not be entirely sufficient either.
> > > >
> > > > Given the updates to the paper and the clarifications by the authors regarding their technique and how it compares to other approaches in the literature, I am updating my score with a recommendation to accept. I think the paper presents a certainly interesting point of view for representation learning that benefits from task information, along with some good discussion on how to best collect data to learn good representations. I think the experimental evaluation with TIA, DREAMER (or similar variants) etc. can be improved further through more rigorous analysis, and for example, applying TIA to several tasks as opposed to only DDMControl - but the initial comparison is still valuable. I believe this paper can contribute to some interesting discussion in the community.

---

> > > ### Author Response · Authors · 2021-11-28
> > > **Re: Reviewer Response (1/2)**
> > >
> > > Thank you for your response!
> > >
> > > **TIA performance**: We too were surprised by the downstream performance in the TIA experiment given the good foreground / background separation. We made sure to use the same RL code for all methods, just loading different checkpoints for the pre-trained encoder weights, so it is unlikely that a bug in RL training led to this result. As mentioned in our original reply one hypothesis is that TIA would still mix task-relevant and irrelevant features in the learned representation. To understand this we need to explain in a little more detail how TIA works.
> > >
> > > TIA’s objective predicts future image observations and rewards, very similar to the “Pred-O+S” method we also include in our experimental comparison -- but crucially TIA predicts future frames using a two-stream architecture with two decoders that separately decode parts of the future frames and get added (via a predicted mask) to form the final prediction (the two streams are what generates the foreground / background plots in Figure 10). TIA explicitly adds objectives that encourage that the representation in one of the decoding streams can predict future rewards (i.e. is task-induced) while the other one contains *no* information about future rewards (i.e. only task-irrelevant features). Following the procedure in the original TIA paper (https://arxiv.org/pdf/2106.15612.pdf), we only use the representation from the task-relevant stream for downstream learning in the TIA experiments we report in Figure 4 since it should contain all task-relevant information.
> > >
> > > Following this, one potential explanation for TIA’s downstream performance in our experiments is that it still uses reconstruction as the training objective. Even though the reconstruction is masked, the task-induced stream still needs to contain all the information for reconstructing the pixel-values in the learned foreground mask -- i.e. it needs to include many details about the *visual appearance* of the agent which are not relevant to the task and thus might mix task-relevant and irrelevant features in the learned representation, making downstream learning less efficient. Another hypothesis is that TIA’s task-induced representation could lack some information due to the limited horizon of the reward prediction. In our experiments with value prediction in TARP-V we learned that predicting values with a too low discount factor leads to worse downstream learning performance (note that in the limit of gamma = 0 we would predict one-step rewards similar to TIA). This is because modeling information like the joint velocities of the agent only becomes important for longer-horizon value prediction -- instantaneous rewards in the Distracting DMControl environments can often be predicted just from the position and velocity of the agent’s center of mass.
> > >
> > > That being said, these are only hypotheses and more experimental analysis is required to rigorously determine the cause of TIA’s downstream performance. After all, the original TIA paper did not test the learned representations for task-transfer. But we hope that this answer more clearly explains the point we made in our original reply.

---

> ### Author Response · Authors · 2021-11-23
> **Rebuttal — TaF6 (1/2)**
>
> We appreciate your comprehensive comments. We address your concerns below.
>
>
>
> **Comparison with similar classes of approaches such as Dreamer and TIA?**
>
> Thank you for this suggestion. We now include a comparison to TIA (see Figure 4(a)) which jointly learns to reconstruct and model task-relevant aspects via reward prediction. In our evaluation we find that TIA successfully disentangles foreground and background in Distracting DMControl (see Figure 10) but still does not lead to effective learning on the downstream task. A possible explanation is that task-irrelevant agent appearance information could still be encoded in the task-induced representation due to the reconstruction objective, slowing the target task learning.
> For comparison with Dreamer, our original evaluation contains a comparison to an approach that trains a representation by multi-task reward prediction and reconstruction (“Pred-O+R”). We now extend this to include experiments where we further finetune this representation on the downstream task (see Appendix C and Figure 11), thus yielding a Dreamer-like representation learning approach that uses model-free RL for downstream task learning instead of Dreamer’s model-based updates. We chose this option instead of running full, model-based Dreamer since we are mainly interested in comparing the quality of the learned representations, and thus used the same downstream task RL algorithm for all representation learning methods. However, we agree with the reviewer that testing all learned representations with a model-based RL algorithm like Dreamer on the downstream task is an interesting direction for future work.
>
>
>
> **Why does unsupervised representation learning fail to focus on task-relevant parts during policy learning? Shouldn’t the representation capture them?**
>
> We agree with the reviewer that the representations learned via unsupervised methods will capture *both* task-relevant and task-irrelevant features in a distributed latent representation. Thus, a downstream policy needs to retrieve the task-relevant information, but also needs to learn to ignore all the irrelevant information, making the learning problem more complex. As illustrated in Figure 6a, we show that unsupervised models can indeed infer task-relevant information like the walker’s joint states, i.e. the task-relevant information is encoded in the representation, but the slower downstream task learning in Figure 4 shows the increased difficulty of *using* this representation due to the need to distinguish between task-relevant and task-irrelevant information.
>
> Following the reviewer’s suggestion we now also **include evaluations in which we finetune the pre-trained encoders on the downstream task** (see Appendix C). We find that the same conclusions apply in this setting: task-induced representations still lead to more efficient learning in visually complex environments. Intuitively, while finetuning can help to focus the learned representation on aspects important to the target task, task-induced representations provide a more beneficial starting point after pre-training since they have already learned to ignore many of the task-irrelevant aspects of the environment during pre-training. Nevertheless we believe that these additional evaluations with finetuning of the learned representation strengthen the evaluation in the paper -- thanks for this suggestion!

---

### Official Review · Reviewer_uzb1 · 2021-11-03

**Correctness:** 3
**Technical Novelty And Significance:** 2
**Empirical Novelty And Significance:** 2
**Recommendation:** 6
**Confidence:** 4

**Main Review:**

strengths:
* experiments comparing with reasonable baselines
* paper is clear

weaknesses:
* i am not sure this paper addresses a real problem. In the limit of high empowerment e.g. human, an agent could achieve a very large number of goals and all potential parts of the screen should be relevant. In that sense this approach only makes sense when a precise goal is targeted and the task space in pre-training can be used to cover it.
* salience maps are not convincing to me. I am not sure what they are meant to convey and in which sense the TARP ones are better than other ones. Background / foreground prediction is also not very relevant in this context.

questions:
* for unsupervised representation learning methods are the unsupervised losses still optimized during fine-tuning. For a fair comparison they should be because it is possible and would make the claim of transfer stronger if they still don't get better results than TARP.

**Summary Of The Paper:**

The paper proposes a representation learning approach for multi-task reinforcement learning. The idea is to pre-train a model using multi-task data either with rewards or behavior cloning. Then fine-tune it on a new task hopefully more efficiently. They contrast it to unsupervised representation learning.

Fine-tuning experimentation is performed with reasonable baselines on 3 different visual domains. The authors probe both what the representation looks like and what pretraining tasks give better performance.




**Summary Of The Review:**

The contribution and novelty are both very limited.

---

> ### Author Response · Authors · 2021-11-23
> **Rebuttal — uzb1**
>
> We thank the reviewer for their detailed feedback. We address your remarks below.
>
> **Task-induced approach only makes sense when a precise goal is targeted**
>
> The reviewer’s remark addresses an important point: while task-induced representations are not as generally applicable to *all possible* downstream tasks as e.g. reconstruction-based representations, they strike a trade-off between being applicable to a wide range of tasks and simplifying the learned representations by modeling only a subset of the available information, especially in visually complex scenes. More formally, we define the set of target tasks that task-induced representations are useful for in Section 4 as all those tasks whose task-relevant information is captured in the union of task-relevant aspects of all pre-training tasks (L140-144). Importantly, this set of target tasks can be wide if the task-induced representation is trained from a diverse set of pre-training tasks (eg “all driving tasks”) , thus a precise target task does not need to be known beforehand.
>
> We agree with the reviewer that an agent that aims to solve *all* possible tasks in an environment would need to model *all* available information. But we would argue that there are many agents of practical relevance that do not need to solve all possible tasks and can thus safely ignore parts of the input information to learn a simpler, easy to learn from representation. For example an autonomous car agent can safely ignore the movement of leaves on the roadside trees without compromising performance in any of the tasks it is supposed to learn. Task-induced representations allow to make this trade-off by varying the number and diversity of pre-training tasks (see Figure 7).
>
> **What does the saliency map explain?**
>
> The saliency map visualizes what information each method models in visually complex scenes.
> Task-induced representation learning methods only model task-relevant parts in visually complex scenes, whereas unsupervised learning methods attempt to model both task-relevant and irrelevant parts and often fail to model all due to the visual complexity of the scenes. We show the saliency map (Figure 5) to qualitatively verify this. For the task-induced approach (TARP-BC), we observe higher saliency values around task-relevant parts such as the walker agent in the distracting DM control environment and cars around the agent in the CARLA environment. On the other hand, the saliency values are scattered around the scenes for unsupervised methods because they attempt to model both aforementioned task-relevant and task-irrelevant information like background videos in distracting DM control and sky / buildings in CARLA.
>
> **What does the background/foreground prediction explain?**
>
> To quantitatively verify whether task-induced representations effectively capture task-relevant parts and discard unnecessary information, we tested whether (1) task-relevant information is modeled in the distracting DM control environment by predicting oracle joint states and (2) whether task-irrelevant information is ignored by classifying the ID of the background video in distracting DMControl. As TARP-BC and TARP-V show lower prediction error than the unsupervised learning methods in state prediction tasks, they can effectively learn task-relevant information. Moreover, task-induced representation learning methods show lower classification accuracy than unsupervised methods for the background video, which implies that they are modeling the task-irrelevant background information less. These quantitative results reinforce the qualitative intuition from Figure 5.
>
> As also pointed out by reviewers k7xA, we believe that these experiments verify the intuition that task-induced representation learning focuses on modeling task-relevant parts and discards unnecessary information.
>
> **Finetune the pre-trained encoder on the target task**
>
> Thank you for this suggestion. In our original experiments we kept the pre-trained encoder weights fixed for all methods during downstream task training to cleanly compare the qualities of the pre-trained representations. While prior work reported mixed results for finetuning pre-trained representations on target tasks [1], we agree with the reviewer that an empirical evaluation of this in visually complex environments is an interesting contribution, and we now added experiments where we finetune the learned representations on the target task (see Appendix C and Figure 11). We find that the conclusions from our original experiments also apply in this setting: task-induced representations overall lead to more efficient downstream task learning in visually complex environments. We will include evaluations with encoder finetuning for all environments in the final version of this paper.
>
> Thank you again for your valuable feedback, please let us know if this addresses your concerns!
>
> [1] Yang et al. “Representation Matters: Offline Pretraining for Sequential Decision Making”

---

> > ### Comment · Reviewer_uzb1 · 2021-11-30
> > **Response**
> >
> > Thank you for the responses.
> >
> > My personal experience in multi-task RL makes me somewhat sceptical about this approach. Firstly because it transforms the problem into task design problem i.e. improving results relies on coming up with new tasks which is open ended in my view and secondly, because the comparison with unsupervised learning methods (VAE, contrastive, etc) is somewhat misleading in that unsupervised learning, requires less instrumentation thus *could* be trained with other data than the simulations that can be used for TARP (the experiments only show the same data). These are the main reasons why my review was "lazy". Personal intuitions aside though I can see how for targeted applications this direction may become fruitful so I am going to change my rating.
> >
> > I find Fig 6c a lot more compelling for demonstrating distractor robustness than the saliency maps and the background prediction experiments. It is ok to include them I just don't find them very compelling.
> >
> > I find it quite promising that the finetuning is stable. Increasing the region of stability seems like a good future research direction as it will make the approach easier to apply to more practical tasks.

---

### Official Review · Reviewer_s9b6 · 2021-11-03

**Correctness:** 4
**Technical Novelty And Significance:** 2
**Empirical Novelty And Significance:** 3
**Recommendation:** 6
**Confidence:** 5

**Main Review:**

Strengths of the paper:

- The paper is very well written.
- The authors evaluate the proposed way of learning task-induced representations to various other "task" agnostic ways of learning representations, and show that the proposed method is able to capture task-relevant information.
- The ablations done in section 5.5 are very informative i.e., set of best practices while collecting datasets for learning task-induced representations

Weakness:
-  For transferring information between different tasks, there are normally three different ways (a) by learning representations of the raw data (b) by learning policies/skills (c) by learning world models. In this work, paper compares to the proposed framework with the scenario when the information is transferred by learning representations of the raw data. It would be also useful to compared to methods/benchmarks which learn the representation of skills/dynamic models.
- It may be useful to compare to methods which explicitly capture task relevant information like information bottleneck approaches [1, 2, 3].
- One other drawback is, that the paper only evaluates by modifying the visual perception by adding distractors. Since the paper uses dynamics model, it would probably be also meaningful to see as to what happens when some "irrelevant" dynamics are added during testing.

[1] Information asymmetry in KL-regularized RL, https://arxiv.org/abs/1905.01240
[2] InfoBot, https://arxiv.org/abs/1901.10902
[3] Learning invariant representations for reinforcement learning without reconstruction, https://arxiv.org/abs/2006.10742

After rebuttal:

I've read the rebuttal. The paper makes it clear they are comparing to methods  that directly learn parametric model (either dynamics model or directly transfer policy)  dynamics models from the offline data (Pred-O and Pred-O+R in Figure 4) and others that directly transfer learned behaviors from the offline data (Policy transfer in Figure 4). I keep my original score.


**Summary Of The Paper:**

In this work, authors formalized the problem of task-induced representation learning. It basically involves investigating various different ways for learning representations from the raw data (for ex. offline dataset) containing "privileged" information (task relevant) from many different tasks. The aim is to leverage task relevant information for learning representations that capture task-relevant information.

The authors evaluate their idea on three different suite of environments (a) Distracting DMControl (b) ViZDoom (c) Autonomous Driving. The authors compare the proposed way of learning task-induced representation to various other methods for representation learning like reconstruction based approaches, mutual information based methods, prediction based methods etc. The results show that the proposed way of learning task-induced representations capture task-relevant information.

**Summary Of The Review:**

The reviewer  found the paper very well written, and tackling an important problem. I like the ablations done in the paper, and the focus on "constructing datasets" for learning task-induced representations.

---

> ### Author Response · Authors · 2021-11-23
> **Rebuttal — s9b6**
>
> Thank you for your thorough review, we hope we can address your concerns below.
>
>
> **Comparison with other approaches that capture task-relevant information?**
>
> Thank you for suggesting several methods which potentially capture task-relevant information. We added DBC [3] as one of the variants of task-induced representation learning in the updated paper. Specifically, We updated section 4 to introduce TARP-Bisim and Figure 4 to show that TARP-Bisim generally performs well compared to other unsupervised learning methods. Furthermore, we add the downstream performance with the representations learned by TIA [6] on distracting DMControl (see Figure 4(a), Figure 10) which learns a dynamics model and leverages reward information to explicitly capture task information.
>
> Information asymmetry in KL-regularized RL [4] and InfoBot [5] tackle online representation learning, whereas our work aims to leverage diverse prior offline datasets for offline representation learnings. Therefore, we think that these methods are not directly applicable in our scenario.
>
>
>
> **Comparison with learning policies/skills and world models?**
>
> Our paper is mainly concerned with comparing the merits of different representation learning approaches in visually complex environments. We agree with the reviewer that there are alternative ways to leverage prior data, for example, via learning dynamics models [1] or direct behavior transfer (e.g. through skills [2]). The comparisons in our paper contain methods that learn dynamics models from offline data (Pred-O and Pred-O+R in Figure 4) and others that directly transfer learned behaviors from the offline data (Policy transfer in Figure 4). However, we agree that further comparisons with the methods mentioned by the reviewer [1,2] can be interesting. While we were unable to complete evaluation of these methods within the limited time of the rebuttal, we plan to include them in the final version of this submission.
>
>
> We again thank you for your valuable feedback, please let us know if this addresses your concerns!
>
> [1] Hafner et al. “Dream to Control: Learning Behaviors by Latent Imagination” \
> [2] Pertsch et al. “Accelerating Reinforcement Learning with Learned Skill Priors” \
> [3] Zhang et al. “Learning Invariant Representations for Reinforcement Learning without Reconstruction” \
> [4] Alexandre et al. “Information asymmetry in KL-regularized RL” \
> [5] Goyal et al. “InfoBot: Transfer and Exploration via the Information Bottleneck” \
> [6] Fu et al. “Learning Task Informed Abstractions”

---

### Decision · Program_Chairs · 2022-01-20

**Decision:**

Accept (Poster)

**Comment:**

The paper is an interesting take on representation learning, using (prior) tasks to determine which information is important. The problem setting is somewhat difficult to pin down, so that that finding the correct comparisons is not obvious and opinions differ on many details of the setup. However, this is not a fault of the paper; it is a general problem the further one moves away from clean settings like classical supervised learning.

There was a lengthy and detailed back-and-forth between the authors and reviewers, where the authors clarified most of the points raised, extended their results, resulting in one reviewer switching from reject to accept.